# Using a Xenogeneic Acellular Dermal Matrix Membrane to Enhance the Reparability of Bone Marrow Mesenchymal Stem Cells for Cartilage Injury

**DOI:** 10.3390/bioengineering10080916

**Published:** 2023-08-02

**Authors:** Weili Shi, Qingyang Meng, Xiaoqing Hu, Jin Cheng, Zhenxing Shao, Yuping Yang, Yingfang Ao

**Affiliations:** 1Department of Sports Medicine, Peking University Third Hospital, Institute of Sports Medicine of Peking University, Beijing Key Laboratory of Sports Injuries, Beijing 100191, China; shiweilixmu@bjmu.edu.cn (W.S.); mengqingyang@bjmu.edu.cn (Q.M.); huxiaoqingbd01@sina.com (X.H.); rubyeyre@163.com (J.C.); shaozhenxing@bjmu.edu.cn (Z.S.); 2State Key Laboratory of Biochemical Engineering, Institute of Process Engineering, Chinese Academy of Sciences, Beijing 100190, China

**Keywords:** tissue engineering, cartilage repair, acellular dermal matrix, microfracture, BMSCs

## Abstract

Due to its avascular organization and low mitotic ability, articular cartilage possesses limited intrinsic regenerative capabilities. The aim of this study is to achieve one-step cartilage repair in situ via combining bone marrow stem cells (BMSCs) with a xenogeneic Acellular dermal matrix (ADM) membrane. The ADM membranes were harvested from Sprague-Dawley (SD) rats through standard decellularization procedures. The characterization of the scaffolds was measured, including the morphology and physical properties of the ADM membrane. The in vitro experiments included the cell distribution, chondrogenic matrix quantification, and viability evaluation of the scaffolds. Adult male New Zealand white rabbits were used for the in vivo evaluation. Isolated microfracture was performed in the control (MF group) in the left knee and the tested ADM group was included as an experimental group when an ADM scaffold was implanted through matching with the defect after microfracture in the right knee. At 6, 12, and 24 weeks post-surgery, the rabbits were sacrificed for further research. The ADM could adsorb water and had excellent porosity. The bone marrow stem cells (BMSCs) grew well when seeded on the ADM scaffold, demonstrating a characteristic spindle-shaped morphology. The ADM group exhibited an excellent proliferative capacity as well as the cartilaginous matrix and collagen production of the BMSCs. In the rabbit model, the ADM group showed earlier filling, more hyaline-like neo-tissue formation, and better interfacial integration between the defects and normal cartilage compared with the microfracture (MF) group at 6, 12, and 24 weeks post-surgery. In addition, neither intra-articular inflammation nor a rejection reaction was observed after the implantation of the ADM scaffold. This study provides a promising biomaterial-based strategy for cartilage repair and is worth further investigation in large animal models.

## 1. Introduction

Articular cartilage serves as a load-bearing and self-lubricated tissue, ensuring frictionless joint movement [1]. Despite possessing remarkable biomechanical properties, articular cartilage injuries are still extremely common due to acute trauma or long-term overuse [2]. Once damaged, the tissue loses its mechanical integrity and the joint fails to translocate smoothly, resulting in increased cartilage erosion and, ultimately, degeneration [1,3]. However, because of its avascular nature and low mitotic ability, articular cartilage has poor self-healing potential [3]. To date, many methods (involving chondroplasty, microfracture (MF), and autologous chondrocyte transplantation) have been used to treat cartilage defects [4,5]. Nevertheless, these cartilage repair therapies have some shortcomings, such as requiring two separate surgical procedures, restoring fibrocartilage, or integrating poorly with the surrounding native tissue [2]. To resolve these difficulties, a promising method has recently emerged to achieve one-step cartilage repair using tissue engineering techniques. This involves the combination of the appropriate biomaterials with endogenous BMSCs [6,7,8].

Biomaterials, as scaffolds, are used to retain more of the early BMSC-rich blood in the defect and to maintain cells in situ until the maturation of the newly repaired tissue by achieving adequate mechanical stability of the early clot. Furthermore, scaffolds can create a supportive microenvironment that promotes cell proliferation, differentiation, and matrix production. They also facilitate the development of tissue with a histological appearance resembling hyaline cartilage. Thereafter, when the scaffold has fulfilled its mission, it should degrade on its own [9,10]. Numerous synthetic or natural biomaterials have been developed for cartilage repair, with studies exploring the introduction of bioactive molecules or functional structures to enhance cell recruitment and proliferation [6,7,11]. Among these approaches, the use of extracellular matrix (ECM) materials holds particular promise, as the optimal environment offered by the ECM is believed to be essential for stem cell differentiation [12,13]. 

The current decellularization approaches for cartilage repair focus on the acellular cartilaginous matrix (ACM) or ECM derived from stem cells or chondrocytes [11,14]. However, there may be concerns regarding the use of a cartilage-specific matrix, particularly due to the lack of standardized protocols [15]. Employing a non-cartilage-specific matrix presents several advantages, including the utilization of standardized protocols that are already established, as well as its easy accessibility and availability in larger volumes. For example, the mesenchymal stem cell (MSC)-based allogeneic acellular dermal matrix (ADM) has been successfully used to regenerate monkey cartilage defects [16]. The ADM membrane is highly accessible and easy to manufacture, with wide resources and low costs. Another study demonstrated that the ADM could regenerate more stable and matured cartilage tissue compared with the ACM, which was confirmed by in vitro and in vivo experiments [17]. However, these studies all required co-culture of cells and scaffolds in vitro, which may cause cell de-differentiation and require two surgeries.

In the present study, we developed a xenogeneic membrane from rat tail-derived ADM material and applied it to cover the defect region after MF in rabbit models. A one-step strategy was employed for the articular cartilage repair, utilizing BMSCs to infiltrate the cartilage defect through the MF. Concurrently, the ADM scaffolds reconstructed a three-dimensional (3D) microenvironment, recruiting a sufficient number of BMSCs to stimulate the regeneration of hyaline-like cartilage. The rat tail-derived ADM was hypothesized to preserve blood clots containing BMSCs, acting as both a physical barrier and a source of biological cues to enhance the repair of cartilage after MF. Initially, the physicochemical properties of the xenogeneic ADM membrane were analyzed, followed by an evaluation of its effectiveness in repairing cartilage defects in a rabbit model. 

## 2. Materials and Methods

The experiments were performed under a project license (No. LA2020020) granted by the ethics committee of Peking University Third Hospital. The animal procedures adhered to the guidelines outlined in the Care and Use of Laboratory Animals (National Academies Press, National Institutes of Health Publication No. 85-23, revised 1996).

### 2.1. Preparation of the ADM Membrane

Dermal matrix membrane was harvested from Sprague-Dawley (SD) rats weighing 200 g. Rat skin from the tail was harvested in full thickness immediately after euthanasia, which was then cut into 1.5 × 2 cm rectangles for preparation of the rat tail-derived ADM. Despite the possibility of affecting the ECM composition or bioactivity, the techniques of decellularization are effective in removing all cell components and DNA. In order to better remove the immunogenic and cell components, we used the techniques of decellularization, which are often used in preparation of ADM membrane [17,18,19]. All samples underwent a soaking process in distilled water for 1 to 3 h, followed by delamination to remove subcutaneous fat, connective tissue, and the epidermis. The dermis sections underwent a decellularization protocol using 0.25% Trypsin/1% Triton X-100. The sections were incubated on a vortex shaker at room temperature with the following solutions: 0.25% trypsin for 6 h; deionized water for 15 min (repeated three times), 3% H_2_O_2_ for 15 min; deionized water for 15 min (repeated twice), 1% Triton X-100 in 0.26% EDTA/0.69% Tris for 6 h and then overnight; deionized water for 15 min (repeated three times), 0.1% peracetic acid/4% ethanol for 2 h; PBS for 15 min; and finally, deionized water for 15 min (repeated twice) [20]. After decellularization, the ADM scaffolds were cut into round sheets (4 mm in diameter). Subsequently, the scaffolds were freeze-dried and sealed before sterilization using cobalt-60 for 24 h and stored at −80 °C in preparation for use.

### 2.2. Assessment of Cellular Content

The acellular content of the ADM samples was assessed using the following criteria: (1) A lack of visible nuclear material in tissue sections stained with hematoxylin and eosin (H&E); (2) Hoechst 33258 working assay for the quantification of double-stranded DNA, less than 50 ng dsDNA per mg ADM dry weight [21]. In addition, the content of ADM collagen was revealed through staining with picrosirius red and examination using polarized light microscopy [22].

### 2.3. Characterization of the Scaffolds

Morphology: The surface and internal cross-section of the scaffolds were examined under scanning electron microscopy (SEM). The scaffolds were freeze-dried and sputter-coated with gold via a Gatan Model 691PIPS (Gatan, Pleasanton, CA, USA). Images of the surface and internal morphologies of the scaffolds were gathered using a FEI Quanta 200F SEM (FEI, Eindhoven, The Netherlands) at 15 KV accelerating voltage.

Physical Properties of the ADM Membrane: The physical properties, consisting of the thickness, diameter, equilibrium swelling ratio (ESR), and porosity of the ADM membrane, were measured. To match the implanted scaffold in vivo, we prepared a circle-shaped ADM membrane with 4 mm diameter. The thickness and diameter of the circle-shaped ADM membrane were measured using a thickness-measuring instrument and Vernier calipers, respectively [7]. To calculate the ESR, the samples (*n* = 5) were immersed in phosphate-buffered saline (PBS) at 37 °C. After 24 h, the samples were removed and weighed using a microbalance after removing any excess surface water with filter papers. The Equilibrium Swelling Ratio (ESR) was calculated using the formula: ESR = (Ws − Wd)/Wd, where Ws represents the weight of the scaffolds in the swollen state and Wd represents the weight of the scaffolds in the dry state. The porosity was calculated as follows: Porosity = Vw/(Vw + Va), where Vw and Va are the volume of the absorbed water and the ADM scaffold, respectively.

### 2.4. In Vitro Experiment

BMSC culture and seeding on scaffolds: BMSCs were harvested from bone marrow aspirates obtained from the distal femur of the aforementioned SD rats and identified based on a previous report [6]. Briefly, the aspirates were cultured in a medium consisting of 89% minimum essential medium α (MEM-α), 10% fetal bovine serum (Hyclone, Logan, UT, USA), and 1% (*v*/*v*) penicillin-streptomycin (Gibco, Grand Island, NY, USA), and then incubated at 37 °C with 5% CO_2_. After 3 days of incubation, the non-adherent cells were discarded, and the remainder were cultured by replacing the medium. When cell confluence was reached, the cells were designated as passage 0, and BMSCs at passage 3 (P3) were utilized for this study [6]. A suspension of P3 BMSCs (50 μL, cell density 8 × 10^6^ cells/mL; a total of 4 × 10^5^ cells) was seeded onto the ADM scaffolds in 12-well tissue culture plates and incubated at 37 °C for 2 h. Subsequently, the BMSC-loaded scaffolds were incubated with 1 mL of complete MEM-α for proliferation or chondrogenic differentiation medium (RASMX-90041; Cyagen Biosciences Inc., Guangzhou, China) for chondrogenesis. After cultivation, the cell-laden scaffolds were assessed at specific time points.

Cell distribution and viability evaluation in scaffolds: The distribution and morphology of the BMSCs seeded on the ADM scaffolds were observed under a confocal microscope (Leica, Nussloch, Germany) according to a previous study [6]. Briefly, after 48 h, the BMSC-loaded scaffolds were washed with PBS and then fixed with 4% paraformaldehyde. Rhodamine Phalloidin (160 nM; Cytoskeleton Inc., Denver, CO, USA) was applied to stain the cytoskeleton of the BMSCs for 1 h at 37 °C. Following washing, the nuclei were counter-stained with Hoechst33258 (2 μg/mL; Fanbo, Beijing, China) for 10 min [23]. The BMSC viability on the scaffolds was quantified using a Cell Counting Kit-8 assay (CCK-8, Sigma, St. Louis, MO, USA) following the manufacturer’s protocol (*n* = 5). Culture dish controls (cells cultured in the dish) were also assessed. In brief, at 1, 3, 5, and 7 days after the same amount of BMSCs were seeded on the ADMs and culture dish, respectively, 50 μL of CCK-8 solution was added to the medium and cultured for 4 h at 37 °C. The OD value was then measured at 450 nm using a plate reader; subsequently, the OD value at each point was normalized against the average of the first day in each group [24].

Chondrogenic matrix quantification: A chondrogenic differentiation medium for BMSCs was used in the culture from RASMX-90041 (Cyagen Biosciences Inc.). The culture medium was changed every three days and collected for biochemical analysis. After 7, 14, and 21 days, the constructs were subjected to biochemical analyses for their glycosaminoglycan (GAG) and hydroxyproline (HYP) contents. The GAG and HYP contents were quantified using a Varioskan Flash reader (Thermo Fisher Scientific, Waltham, MA, USA). The scaffolds seeded with cells were digested for 24 h in a pre-prepared papain solution (Sigma) at 60 °C overnight for GAG estimation after being weighed using a microbalance. The total sulfated GAG content was measured using the dimethylmethylene blue (DMMB, Sigma) assay, while the collagen content was evaluated by quantifying the HYP content. Subsequently, aliquots of the same digest solution were hydrolyzed in 6 M HCl at 120 °C for 2 h. The resulting hydrolyzed solution was then subjected to a chloramine-T/Ehrlich’s spectrophotometry assay at a wavelength of 560 nm to measure the HYP content. The HYP content was determined based on a standard curve of L-hydroxyproline (Sigma). Both the GAG and HYP contents were normalized by their wet weight [25,26].

### 2.5. In Vivo Experiment

#### 2.5.1. Animal Surgery Procedure

Twelve adult male New Zealand white rabbits weighing 2.5 to 3.0 kg were used in this study. After anesthesia and routine pre-surgery preparation, the knee joint of a rabbit was exposed after the patella was dislocated, and cylindrical full thickness chondral defects (4-mm in diameter) were created on the trochlear groove of the distal femur with a corneal trephine. Rabbit BMSCs were obtained from the marrow blood through standard microfracture, and then the tested ADM scaffold was implanted by matching with the defect in the right knee with the porous surface down (ADM group). Following that, the knee was cycled from flexion to extension to secure the localization of the scaffolds within the defect. In the left knee, isolated microfracture was performed as the control (MF group). Subsequently, the joint was sutured, and prophylactic penicillin was administered intramuscularly to prevent infection. Post-surgery, the rabbits were housed individually in cages, provided with standard food and water, and allowed unrestricted movement. At 6, 12, and 24 weeks postoperatively, three rabbits were sacrificed for subsequent analysis.

#### 2.5.2. Synovial Fluid Analysis and Macrography

At 6, 12, and 24 weeks post-surgery, synovial fluid was obtained bilaterally with a 1 mL syringe and an 18 gauge needle. The collected fluid was then subjected to centrifugation at 4000 rpm for 20 min at 4 °C. Subsequently, the supernatants were obtained and frozen at −80 °C. Enzyme-linked immunosorbent assays (ELISAs) were employed to analyze the inflammatory factors interleukin-1 (IL-1) and tumor necrosis factor-alpha (TNFα) in the synovial fluid. The ELISA kits used for the analysis were the Rabbit IL-1 ELISA Kit (I079SC) and Rabbit TNFα ELISA Kit (T103SC), obtained from Hermes Criterion Biotechnology based in Vancouver, Canada. Additionally, filling of the defects, interfacial integration, and surface smoothness of each repaired tissue at different time points and groups were evaluated through comprehensive observation. 

#### 2.5.3. Histological Assessment of Repaired Tissue

After the whole observation, histological specimens were washed with PBS, fixed in 4% paraformaldehyde (pH 7.4) for 48 h at 4 °C, decalcified in 20% EDTA (pH 7.2) for about one week, dehydrated in a graded ethanol series, and embedded in paraffin. Coronal sections (5-μm thick) were then cut through the center of the operative site and stained with hematoxylin and eosin (H&E), Safranin-O, and immunostaining using a type II collagen antibody (Novabiochem, Burlington, MA, USA) according to standard protocols. The menisci were also stained to evaluate the abrasion within the joints. The International Cartilage Repair Society (ICRS) macroscopic evaluation scale was utilized to assess the degree of cartilage repair in different groups [27,28]. The key parameters evaluated included the degree of cartilage defect, interfacial integration of the border zone, macroscopic appearance, and overall repair assessment [27]. The level of articular cartilage repair in the different groups was assessed using a modified O’Driscoll grading system [29,30,31,32], consisting of the nature of the predominant tissue (cellular morphology and Safranin-O staining of the matrix), structural characteristics (surface regularity, integrity, thickness, and bonding to adjacent cartilage), and absence of cellular changes indicating degeneration (hypocellularity, chondrocyte clustering, and absence of degenerative changes in adjacent cartilage) [33,34]. All specimens were independently evaluated by two professionals. The evaluators were blinded to the treatment during the assessments [34].

#### 2.5.4. Nanoindentation Assessment

Biomechanical analysis of the repaired tissue was conducted using nanoindentation, as described previously [6,35]. Five samples were obtained from the central portion of the repaired tissues, while another five control samples were obtained from the non-operated normal trochlea of the knee. Hydration was maintained using a circumfluent PBS solution at room temperature. Nanoindentation was conducted using a TriboIndenter (Hysitron Inc., Minneapolis, MN, USA) equipped with a conospherical diamond probe tip featuring a 400-mm radius curvature. A trapezoidal load function, comprising loading (10 s), hold (2 s), and unloading (10 s), was applied at each indent site. The indentations were force controlled to a maximum depth of 500 nm. The micro scanning apparatus was used to capture the microscopic geomorphology of the indentation zones [6].

### 2.6. Statistical Analysis

Data analysis was performed using the SPSS 22.0 software (IBM Corp., Armonk, NY, USA). The results were presented as mean ± standard deviation (SD). Differences among groups were assessed using a one-way ANOVA analysis after a test of the homogeneity of variances. Student’s *t*-test was used to evaluate within-group data, with a statistical significance set at *p* < 0.05. 

## 3. Results

### 3.1. Gross Observation and SEM Images of ADM Scaffolds

A polar structure with a porous surface and an impermeable surface of the ADM membrane could be clearly determined under macroscopic observation (Figure 1A). Under SEM, the perforated surface of the ADM membrane showed many clustered pores, while the imperforate surface showed a compact structure that appeared very smooth, without any pores. The cross section of the ADM membrane displayed a rough morphology, characterized by irregularly arranged collagen fibers and interspersed pores (Figure 1B). 

### 3.2. Evaluation of ADM

No nuclei were observed in the ADM assessed through H&E stained sections compared with the normal dermal matrix (Figure 2A). The samples had <50 ng of dsDNA per mg initial dry weigh, as measured with the Hoechst 33258 (Figure 2B). In addition, we observed the pore structure from the H&E imaging and collagen composition under polarized light microscopy (Figure 2C). The decellularized protocol was proven to be effective based upon the above results [21].

### 3.3. Physicochemical Properties of the ADM Scaffold

As summarized in Table 1, the diameter and thickness of the ADM, which was cut into a circular shape in the final step, were 3.92 ± 0.21 mm and 0.98 ± 0.19 mm, respectively. The results of the ESR showed that the ADM can absorb water. This excellent porosity indicated that the scaffold could provide a favorable microenvironment for cell growth.

### 3.4. Distribution, Viability, and Chondrogenesis of BMSCs Seeded on ADM Scaffolds

Confocal microscopy was employed to evaluate the adhesion and morphology of the BMSCs cultured on the ADM scaffolds. After 48 h of culture, the BMSCs demonstrated robust growth. The typical spindle-shaped BMSC morphology was demonstrated using cytoskeleton immunostaining images (Figure 3A). Although statistically significant differences were observed between the two groups at days 3 and 5, there was no significant difference at day 7 regarding cell proliferation. The ADM scaffold demonstrated no significant impact on the BMSC viability compared with the culture dish commonly used for cell culture. Moreover, the cell proliferation remained significant from day 1 to day 7 in the ADM group (Figure 3B). 

The levels of GAG and HYP (indicating the collagen content), which were both normalized by their wet weight, were assessed to quantify the cartilaginous matrix production by the BMSCs within the ADM scaffolds. There was a significant increase in the GAG content in the ADM scaffold. Similar to the results of GAG, the BMSCs within the scaffolds produced significantly more collagen over time (Figure 3C).

### 3.5. Abrasion and Inflammation Detection of the Repaired Knees

The abrasion and inflammation were evaluated during the cartilage repair at different times. The medial menisci in the MF and ADM groups were analyzed using H&E staining, and the structure and morphology of the menisci in both groups were unbroken at 24 weeks (Figure 4A). The levels of inflammatory factors (IL-1 and TNFα) within the joint fluid in the ADM group were maintained at a relatively low level after 6 weeks, with no statistically significant difference compared with those of the MF control group (Figure 4B,C). These results demonstrated that no inflammation or rejection reaction was induced by the implantation of the ADM scaffold.

### 3.6. Gross Observation and Histological Evaluation of Cartilage Repair

At 6 weeks post-surgery, based on visual inspections, the defect sites of the two groups were packed with some white tissues and the repaired tissues were lower than the surrounding cartilage. Nearly half of the defects were packed with brown-colored tissues in the MF group (Figure 5). Based on the H&E staining, the non-uniform repaired tissue that filled in the defect was not as smooth as that of the native cartilage, and the interface between the normal cartilage and the regenerated tissue was apparent in both groups. The repair tissues observed in the MF group exhibited reduced thickness and integration compared to those in the ADM group (Figure 6).

At 12 weeks after surgery, visual inspections revealed that the defects in the MF group exhibited shallower depths and were filled with rough fiber-like tissue. The presence of brown-colored tissues was still observed within the repair tissues (Figure 5). In contrast, the defects in the ADM group were completely filled, despite some disintegration with the adjacent normal cartilage (Figure 5). The repair tissue in the ADM group appeared thinner when compared to the surrounding normal cartilage. However, the H&E staining indicated that the repaired tissue in the ADM group was undergoing remodeling, displaying interfacial integration between the defects and normal cartilage (Figure 6). 

At 24 weeks post-surgery, the defects in the MF group were filled with fibrous tissue, and the adjacent cartilage was degenerative (Figure 5 and Figure 6). The filling of the defects in the ADM group was consistent, smooth, and newly produced cartilage repair, without significant disintegration (Figure 5 and Figure 6). These results showed that ADM +MF could accelerate the regeneration and remodeling of articular cartilage defects.

### 3.7. Cartilage-Specific Staining

Safranin-O and immunohistochemical staining for sulfated GAGs and type II collagen were used to assess the quality of the cartilage repair. At 6 weeks, the repaired tissues of the MF group showed no Safranin-O staining, whereas two of the three samples in the ADM group showed lighter staining compared with the normal cartilage samples. The same phenomenon was found at 12 weeks in both the MF and ADM groups. At 24 weeks, two of the three repaired tissues in the MF group showed lighter staining compared with the normal cartilage samples, whereas the ADM group showed uniform Safranin-O staining that was much closer to the normal cartilage in all three samples (Figure 7). 

Similar to the Safranin-O staining, the immunohistochemical staining for type II collagen showed significantly stronger expression in the ADM group compared to the MF group at 6, 12, and 24 weeks after surgery. This indicates the superior cartilage repair quality and quantity in the ADM group without hypertrophic cartilage remodeling, as opposed to the MF controls (Figure 8).

The ICRS macroscopic scores and modified O’Driscoll grading system were statistically analyzed for each group (Figure 9). The results demonstrated that the ADM group outperformed the MF group in terms of the scores. In conclusion, the ADM group exhibited significantly higher macroscopic and histological scores at each evaluation point, indicating superior outcomes compared to the MF group (*p* < 0.05).

### 3.8. Biomechanical Properties of the Repaired Cartilage

At 24 weeks post-surgery, nanoindentation was conducted to evaluate the biomechanical properties of the repaired cartilage zones. Using micro-scanning, the results showed that the articular surface of the normal cartilage was smooth. As the microscopic appearance showed, the surface of the neo-tissue in the MF control group was rougher than that in the ADM group. The surface of the cartilage repaired by the ADM was more similar to the normal cartilage (Figure 10A). Similar to the native cartilage, the tissue in the ADM group exhibited a significantly higher reduced modulus compared to the MF group (Figure 10B). In addition, the cartilage repaired by the ADM was harder than that repaired by the MF (Figure 10C).

## 4. Discussion

Clinically, MF is commonly utilized as a primary treatment for articular cartilage injury due to its simplicity and minimally invasive nature. However, this approach often results in a higher incidence of fibrocartilage formation and inadequate integration with the native articular cartilage, adversely affecting the long-term outcomes. First, inadequate repair can be attributed to a scarcity of chondrogenic cells originating from the subchondral bone marrow and their limited ability to migrate to the defect site. Second, there was no mechanical support or suitable microenvironment for the proliferation and differentiation of the BMSCs, which is indispensable during the formation of the clot and neo-cartilage [36]. In this study, we explored the use of an ADM scaffold for cartilage repair. One surface of the ADM membrane appeared smooth and compact, while the opposite surface displayed a rough morphology with collagen fibers and interspersed pores. The impermeable surface can retain more MSC-rich blood and provide protection of the early clot not washed by joint fluid after MF in vivo as a barrier. The perforated surface with a suitable pore size and high porosity can provide a 3D microenvironment for cell proliferation, differentiation, and ECM production, which were all positive indicators for inducing increased hyaline cartilage [13]. Cartilage relies on diffusion to secure nutrients and growth factors to chondrocytes because of its avascularity. Thus, the ADM membrane not only acts as a barrier, but its permeability for the slow diffusion of soluble factors, via its wettable features, was also proven by the results of the ESR. In addition, the ADM is thin, flexible, tough, and possesses a good water absorption capacity.

In recent decades, acellular materials have been developed to enhance tissue regeneration and functional recovery because acellular ECMs provide critical biological cues and genuine 3D microstructures for cell adhesion, proliferation, and differentiation [15,37]. ADM, as one such acellular material, has been used in clinical surgical applications, such as abdominal wall reconstruction, breast reconstruction, and anterior tracheal reconstruction. It has been reported that allogeneic ADM can result in neo-cartilage formation similar to native cartilage [16,38]. It was thought that ADM is highly accessible and easy to manufacture, with wide resources and low costs, compared to other acellular materials, such as ACM and acellular bone matrix (ABM) [8,17,39]. The use of allograft or exogenous cells in this research might have limited their clinical applications in the future. In the present study, we used a xenogeneic ADM derived from rat tails to repair rabbit cartilage, which broadened the material sources. Theoretically, as the cellular components—as the major sources of immunogens—are substantially eliminated, a xenogeneic ADM would cause no additional immune response. By detecting inflammatory factors within the joint fluid in the ADM group in vivo, we demonstrated that the ADM scaffold did not induce additional inflammation or a rejection reaction. Macroscopically, no signs of joint erosion, inflammation, swelling, or deformity were observed in the ADM group.

Thus, we designed a new method of neo-cartilage induction and formation by releasing autologous BMSCs combined with using the ADM scaffold. This method could achieve the one-step repair of an articular cartilage injury without secondary surgery or the use of endogenous BMSCs and without donor site morbidity. Preliminary studies have confirmed that this method is feasible and has excellent effects for cartilage repair [6,7,8].

The collagen content and distribution of the scaffold were essential in the differentiation of the BMSCs into chondrocytes [40]. Abundant and structured collagen fibers could be observed in the ADM scaffold through special picrosirius red staining when evaluated using polarized light [22]. The BMSCs exhibited favorable growth on the ADM scaffolds, maintaining their typical spindle-shaped morphology, as observed through confocal laser microscopy in vitro. The ADM scaffold did not have a notable effect on the BMSC viability, as indicated by the CCK-8 assay. The production of GAG and collagen, which are components of the cartilage ECM, are important for cartilage regeneration [15]. The in vitro experiment demonstrated that the levels of GAG and HYP increased significantly over time. It was reported that cartilage repair involves multiple cells and factors, and that the ADM may have some possible effects on them [41,42]. The ADM is harvested from dermis and primarily consists of collagen I and III, exhibiting low immune reactions and an inhibitive effect in inflammatory cytokines in vivo. Previous studies have demonstrated that the ADM promotes stable and homogeneous cartilage formation. The homogeneous distribution of chondrocytes within the ADM scaffold enhances the quality of regenerated cartilage, resulting in superior neocartilage [17]. These results indicated a superior capacity of chondrogenic differentiation of the BMSCs cultured within the ADM scaffold.

Although it was unclear whether the scaffold-alone treatments showed more favorable efficacy for cartilage repair, the ADM membrane was proven to be beneficial for the repair of cartilage defects after MF. The utilization of the scaffold platform facilitated sufficient defect filling, surpassing the outcome of MF alone. Furthermore, the ADM group demonstrated improved interfacial integration between the repaired cartilage and the surrounding tissue, as well as upgraded biomechanical properties, compared to the MF group. These observations collectively suggested that the ADM scaffold enhanced both the biological and physical characteristics of the resulting blood clot. By promoting cell retention and providing a conducive microenvironment for chondrogenesis, the ADM scaffold contributed to these improvements. The image analysis of the Safranin-O and immunohistochemical staining reflecting GAG and Type II collagen expression offered key information regarding the quantity and quality of the repaired cartilage. The expression of GAG and Type II collagen were substantially increased in the ADM group compared with those in the MF group, suggesting that the ADM scaffold performed well and produced more hyaline-like cartilage. It has been reported that ADM contains no type II collagen; therefore, the type II collagen in the repaired cartilage was generated by differentiated endogenous cells. The deficiency of this experiment is the lack of comparative results with allogeneic ADM and other biomaterials.

## 5. Conclusions

In conclusion, we proposed a promising method to achieve one-step cartilage repair by combining a xenogeneic ADM membrane derived from rat tail with endogenic BMSCs in a rabbit model. The ADM is abundant, multiple, thin, flexible, tough, and possesses good water absorption capacity. The ADM scaffold provided a suitable microenvironment for cartilage repair and could improve the cartilage repair quality compared with MF alone, without inducing extra inflammation or a rejection reaction.

## Figures and Tables

**Figure 1 bioengineering-10-00916-f001:**
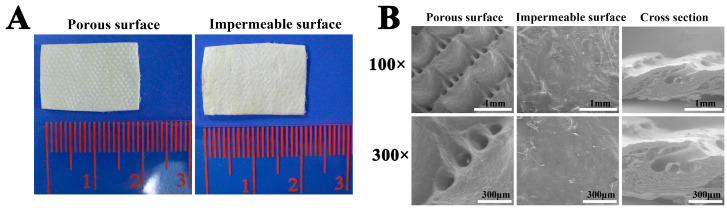
Gross observation and SEM images of ADM Scaffolds. (**A**) Polar structure of the AMD scaffold with porous and impermeable surfaces. (**B**) Polar surface and sectional image of the ADM scaffold under scanning electron microscopy (SEM; 100× and 300×). SEM, scanning electron microscopy; ADM, acellular dermal matrix.

**Figure 2 bioengineering-10-00916-f002:**
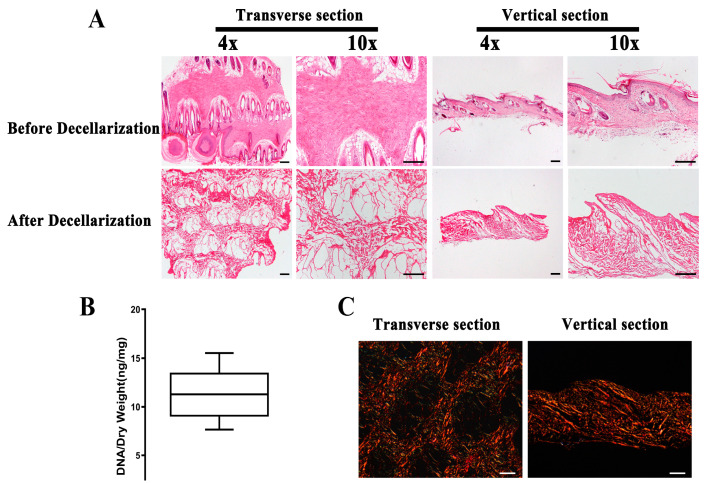
The evaluation of ADM. (**A**) A lack of visible nuclear material in different sections stained with H&E after decellularization. (**B**) Quantification of residual DNA after decellularization of the dermal matrix. (**C**) Evaluation of the organization of the collagen fibers in the ADM under polarized light microscopy. (*n* = 5, scale bar = 200 μm) ADM, acellular dermal matrix.

**Figure 3 bioengineering-10-00916-f003:**
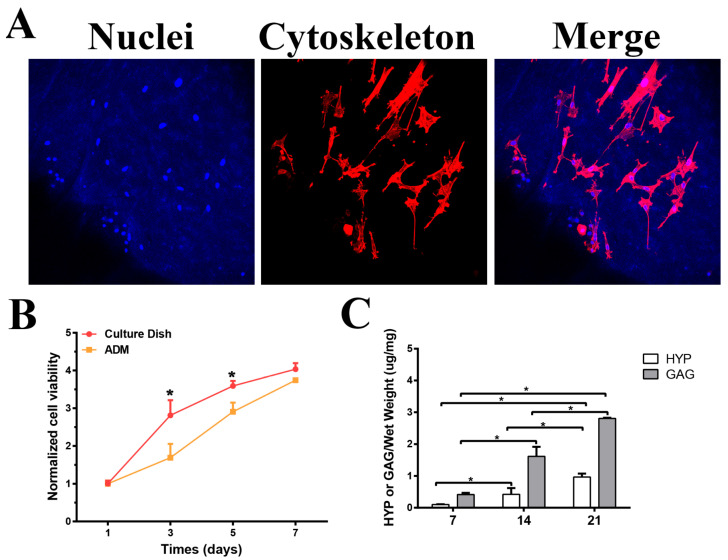
In vitro proliferation and cartilaginous matrix production of BMSCs in ADM scaffolds. (**A**) The Phalloidin/Hoechst assay revealed the typical spindle-shaped morphology of BMSCs after 3 days of incubation with ADMs. (**B**) The viability of BMSCs was assessed using the CCK-8 assay on both ADM scaffolds and culture dishes. (**C**) The production of cartilage-specific matrix in ADM scaffolds was evaluated through HYP assay for collagen quantification and GAG assay for cartilaginous matrix production at multiple time points. (*n* = 3, * *p* < 0.05, scale bar = 200 μm). ADM, acellular dermal matrix; CCK-8, cell counting kit-8 assay; HYP, hydroxyproline; BMSCs, bone marrow stem cells.

**Figure 4 bioengineering-10-00916-f004:**
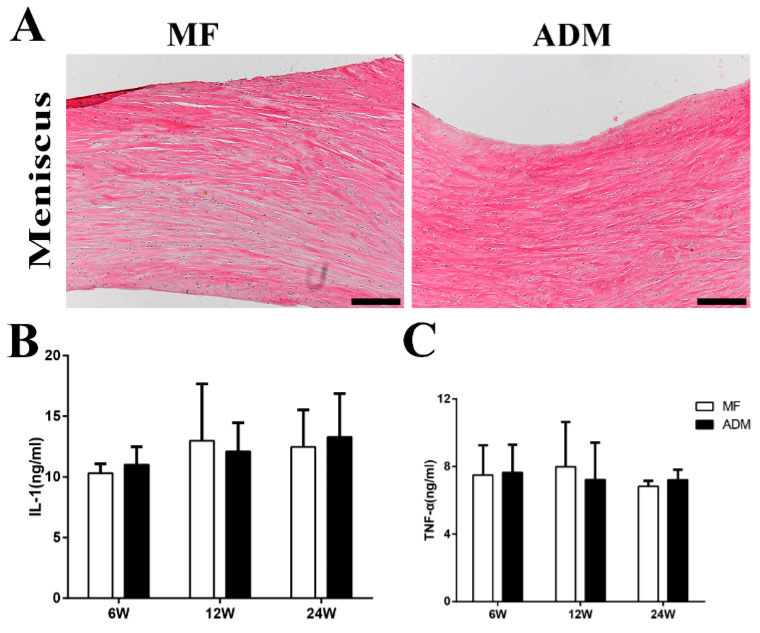
(**A**) The histological assessment of abrasion on load-bearing tissue in the joint included observation of the meniscus, which remained intact in both groups after 24 weeks. (scale bar = 200 μm) (**B**) The levels of Interleukin-1 in joint fluid were measured at 6, 12, and 24 weeks post-surgery. (**C**) The levels of tumor necrosis factor-α in joint fluid were measured at 6, 12, and 24 weeks post-surgery. (*n* = 3) ADM, acellular dermal matrix; MF microfracture.

**Figure 5 bioengineering-10-00916-f005:**
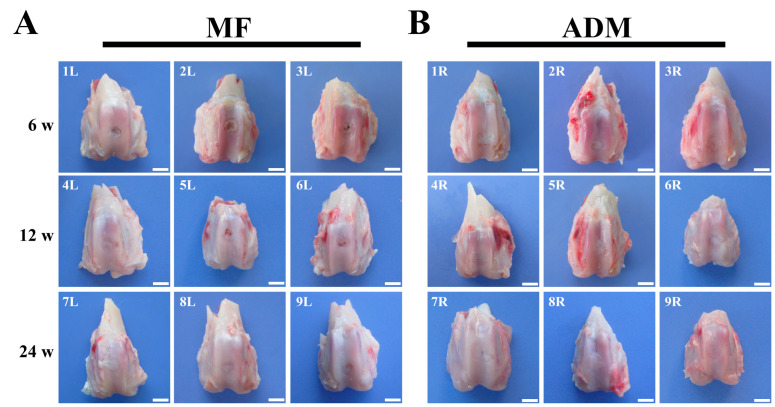
Macroscopic observation of the repaired cartilage at 6, 12, and 24 weeks after MF in a rabbit model with or without ADM scaffolds. (**A**) The cartilage defects were uncovered on the left knee joints (MF group). (**B**) The cartilage defects were covered with ADM scaffold on the right knee joints (ADM group). The rabbits were sacrificed at 6, 12, and 24 weeks post-treatment, and representative images from each group are provided. (L or R indicates left or right knee joints. Scale bars = 4 mm. *n* = 3/group) ADM, acellular dermal matrix; MF microfracture.

**Figure 6 bioengineering-10-00916-f006:**
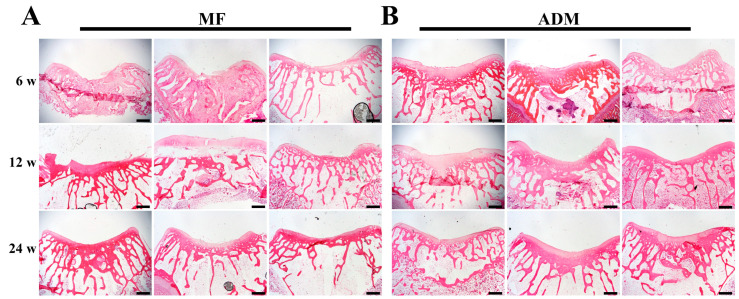
Histological assessment of repaired cartilage in vivo. (**A**) H&E staining at 6, 12, and 24 weeks in the MF group. (**B**) H&E staining at 6, 12, and 24 weeks in the ADM group. (scale bar = 400 μm, *n* = 3/group) ADM, acellular dermal matrix; MF microfracture.

**Figure 7 bioengineering-10-00916-f007:**
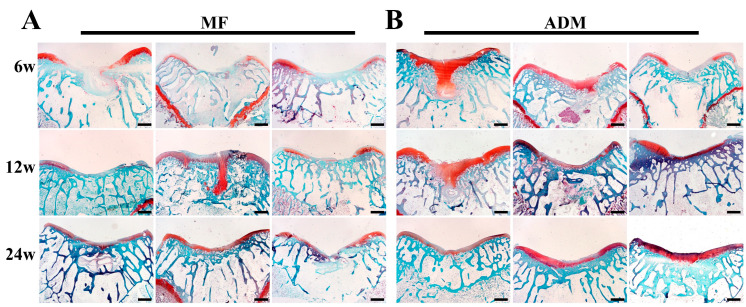
Assessment of GAG contents in repaired cartilage in vivo from all samples. (**A**) Safranin-O staining of repaired cartilage at 6, 12, and 24 weeks in the MF group. (**B**) Safranin-O staining of repaired cartilage at 6, 12, and 24 weeks in the ADM group. (scale bar = 400 μm, *n* = 3/group) ADM, acellular dermal matrix; MF microfracture.

**Figure 8 bioengineering-10-00916-f008:**
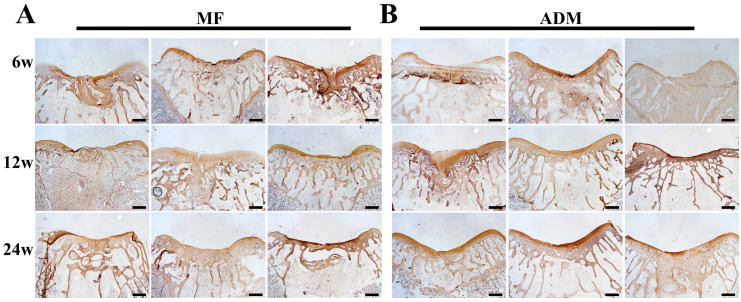
Type II collagen assessment of repaired cartilage in vivo from all samples. (**A**) Immunohistochemical staining of repaired cartilage at 6, 12, and 24 weeks in the MF group. (**B**) Immunohistochemical staining of repaired cartilage at 6, 12, and 24 weeks in the ADM group. (scale bar = 400 μm, *n* = 3/group) ADM, acellular dermal matrix; MF microfracture.

**Figure 9 bioengineering-10-00916-f009:**
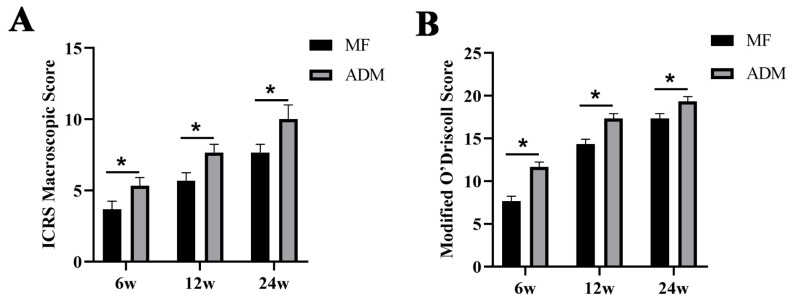
(**A**) Statistical analysis of the International Cartilage Repair Society macroscopic evaluation of cartilage repair (* *p* < 0.05); (**B**) Statistical analysis of the modified O’Driscoll grading system (* *p* < 0.05); ADM, acellular dermal matrix; MF microfracture.

**Figure 10 bioengineering-10-00916-f010:**
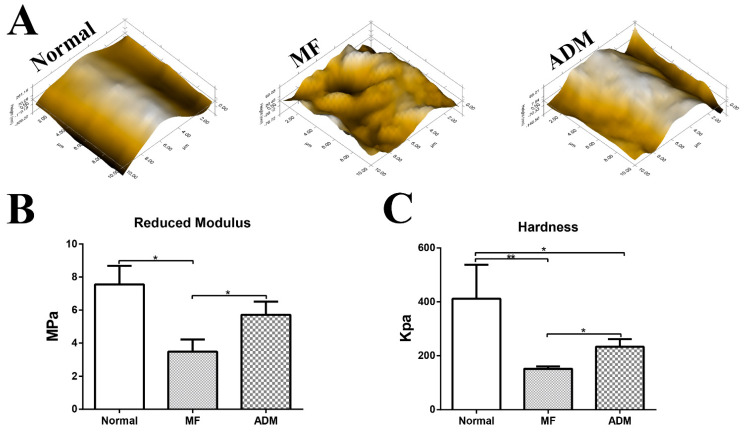
Biomechanical properties of repaired cartilage at 24 weeks. (**A**) Microscopic geomorphology of the repaired zone in different groups was acquired using nanoindentation and compared with normal cartilage. (**B**) Reduced modulus of the repaired tissue in the different groups. (**C**) Hardness of the repaired tissue in different groups. (*n* = 5, * *p* < 0.05, ** *p* < 0.01). ADM, acellular dermal matrix; MF microfracture.

**Table 1 bioengineering-10-00916-t001:** Physicochemical properties of the ADM membrane.

Physical and Chemical Properties			
Diameter	3.92 ± 0.21 mm	ESR	1.90 ± 0.14%
Thickness	0.98 ± 0.19 mm	Porosity	0.59 ± 0.06%

All data were presented as the mean ± standard deviation (SD).

## Data Availability

The data presented in this study are available on request from the corresponding author.

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
