# Peer review of "Using a Xenogeneic Acellular Dermal Matrix Membrane to Enhance the Reparability of Bone Marrow Mesenchymal Stem Cells for Cartilage Injury"

_bioengineering, 2023, doi:10.3390/bioengineering10080916_

Round 1
Reviewer 1 Report
This paper presents an intriguing study on the utilization of Xenogeneic Acellular Dermal Matrix Membrane to enhance the reparability of bone marrow mesenchymal stem cells for cartilage injury. However, some points need further clarification.
1- it would be beneficial to elaborate on the rationale behind employing the dermal matrix membrane.
2- Additionally, including details about the chondrogenic induction medium used would enhance the comprehensiveness of the study.
3- Lastly, it is recommended to provide higher quality images for Figure 2 to improve visual clarity.
1- Sentence 383: "injury" is missing an article before it. It should be "an articular cartilage injury."
2- Sentence 384: "However, it is usually associated with a greater proportion of fibrocartilage formation and poor integration" - "integration" is incomplete, and there seems to be a line break. It should be "However, it is usually associated with a greater proportion of fibrocartilage formation and poor integration."
3- Sentence 386: "partly cause by" - "cause" should be "caused."
4- Sentence 388: "there were no mechanical support" - "were" should be "was" to agree with the singular subject "mechanical support."
Reviewer 2 Report
This study explored acellular dermal matrix membranes (ADM) from Sprague-Dawley rats for cartilage regeneration. The study involved characterizing the ADM membranes to assess their suitability for cell distribution, chondrogenic matrix quantification, and viability evaluation. In vivo experiments comparing ADM scaffold to microfracture in rabbits showed superior tissue formation, integration, and cartilage regeneration without adverse reactions. These findings highlight the potential of xenogeneic ADM membranes for one-step cartilage repair, warranting further investigation in larger animal models. Nonetheless, the authors should address several questions,
1. Figure 2B should include a comparison with the control (before and after decellularization) to provide a comprehensive analysis of the effects of the decellularization process on the ADM membranes.
2. How look like BMSC adhesion rate on ADM?
3. In addition, it is suggested that the authors measure objective values such as ICRS score in Figure 6 and CAG optical density in Figure 7 to compare between the microfracture (MF) and ADM groups. This would provide a more comprehensive evaluation of the effectiveness of the ADM scaffold in cartilage regeneration.
4. How do these findings compare to your previous research (American journal of translational research 2019, 11, 6650-6659)? It would be helpful to include a discussion that highlights the differences between acellular bone matrix (ABM) and ADM.
Reviewer 3 Report
The paper entitled “Using a xenogeneic acellular dermal matrix membrane to enhance the reparability of bone marrow mesenchymal stem cells for cartilage injury” is presented. The authors showed that a one-step cartilage repair in situ by combining bone marrow mesenchymal stem cells with a xenogeneic acellular dermal matrix membrane. This study suggests a prototype of biomaterial for cartilage repair that might have translational potential.
There are some possible issues
Fig. 1 and 2. Can you verify that the acellular dermal matrix is completely absent of the xenogeneic DNA ?
Figure 2. (C) Evaluation of the organization of the collagen fibers in the ADM. Can you provide quantitative analyses?
Figure 3B, can you provide comparative analyses with a significant difference or not ?
Previous studies have found that cartilage repair involves multiple cells and factors (for example, PMID: 26962464; PMID: 31312184). It would be informative to discuss the possible effects of acellular dermal matrix on these multiple cells and factors.
Figures 6 and 7. Histological assessment of repaired cartilage. Can you provide quantitative analyses? with an OARIS scoring system?
Author Response
"Please see the attachment

Round 2
Reviewer 2 Report
The paper can be accepted for publication.
Author Response
Thank you for your recognition

Reviewer 3 Report
This is a revised paper and the authors have addressed questions mostly. It was suggested to discuss the possible effects of acellular dermal matrix on these multiple cells and factors that are involved in cartilage repair and OA (for example, PMID: 26962464; PMID: 31312184). It would be informative to discuss the possible effects of acellular dermal matrix on these multiple cells and factors as mentioned, which would enhance cellular and molecular mechanistic insights.
Author Response
Response: Thanks for your comments. According to your suggestion, we have cited the two mentioned articles (PMID: 26962464 and 31312184) and added some discussion in the revised manuscript as: “It has reported that cartilage repair involves multiple cells and factors and ADM may have some possible effects on them[1,2]. ADM is derived from dermis and mainly consists of collagen I and III, erecting low immune reactions and exhibiting inhibitive effect in inflammatory cytokines and matrix metalloproteinases in vivo. Previous studies have demonstrated that ADM promotes stable and homogeneous cartilage formation. The homogeneous distribution of chondrocytes within the ADM scaffold enhances the quality of regenerated cartilage, resulting in superior neocartilage.[3]”
- Zhang, W.; Ouyang, H.; Dass, C.R.; Xu, J. Current research on pharmacologic and regenerative therapies for osteoarthritis. Bone research 2016, 4, 15040, doi:10.1038/boneres.2015.40.
- Zhang, W.; Robertson, W.B.; Zhao, J.; Chen, W.; Xu, J. Emerging Trend in the Pharmacotherapy of Osteoarthritis. Frontiers in endocrinology 2019, 10, 431, doi:10.3389/fendo.2019.00431.
- Wang, Y.; Xu, Y.; Zhou, G.; Liu, Y.; Cao, Y. Biological Evaluation of Acellular Cartilaginous and Dermal Matrixes as Tissue Engineering Scaffolds for Cartilage Regeneration. Frontiers in cell and developmental biology 2020, 8, 624337, doi:10.3389/fcell.2020.624337.